# The Empirical Mean is Minimax Optimal for Local Glivenko-Cantelli

**Doron Cohen** [1]  **Aryeh Kontorovich** [1]  **Roi Weiss** [2]

## Abstract

We revisit the recently introduced Local Glivenko-Cantelli setting, which studies distribution-dependent uniform convergence rates of the Empirical Mean Estimator (EME). In this work, we investigate generalizations of this setting where arbitrary estimators are allowed rather than just the EME. Can a strictly larger class of measures be learned? Can better risk decay rates be obtained? We provide exhaustive answers to these questions—which are both negative, provided the learner is barred from exploiting some infinite-dimensional pathologies. On the other hand, allowing such exploits does lead to a strictly larger class of learnable measures.

## 1. Introduction

Cohen & Kontorovich (2023) initiated the study of the *local Glivenko-Cantelli* setting: laws of large numbers that are uniform over a function class but rather than being *universal* over all distributions, feature a delicate dependence of the risk decay on the (local) sampling measure. This naturally led to the *binomial empirical process*: for a fixed $p \in [0,1]^{\mathbb{N}}$ and each $n \in \mathbb{N}$, we have a sequence of independent $Y_j \sim \text{Binomial}(n, p_j)$, which are centered and normalized to obtain $\bar{Y}_j := n^{-1} Y_j - p_j$. The object of interest is the expected uniform absolute deviation:

$$\Delta_n := \mathbb{E} \sup_{j \in \mathbb{N}} |\bar{Y}_j|. \quad (1)$$

More generally, one could imagine fixing a distribution $\mu$ on $\{0,1\}^{\mathbb{N}}$, sampling $X^{(1)}, X^{(2)}, \dots, X^{(n)}$ i.i.d. from $\mu$, and estimating $p := \mathbb{E} X^{(1)}$ via the Maximum-Likelihood Estimator (MLE) $\hat{p} := \frac{1}{n} \sum_{i=1}^{n} X^{(i)}$. In the case where $\mu$ is a product measure (that is, the components of the vector

$X \sim \mu$ are mutually independent), $\mathbb{E} \|\hat{p} - p\|_{\infty}$ recovers the expression in (1). Despite its austere appearance, the binomial empirical process with independent coordinates $Y_j$ under $\ell_{\infty}$-norm deviation already captures much of the richness of problem. Extensions to more general product distributions $\mu$ over $[0,1]^{\mathbb{N}}$ are straightforward (Blanchard & Voráček, 2024, Corollary 6) and the behavior under $\ell_r$ norms for $r < \infty$ is considerably simpler (Proposition 7 ibid.). Finally, the in-expectation bounds are readily converted to high-probability tail bounds (Proposition 9 ibid.), and all of the upper bounds stated for product measures hold verbatim for arbitrary correlations.

For the purpose of analyzing (1), Cohen & Kontorovich showed that there is no loss of generality in restricting $p$ to the set $[0, \frac{1}{2}]^{\mathbb{N}}_{\downarrow 0}$, consisting of all $p \in [0, \frac{1}{2}]^{\mathbb{N}}$ with $p_j \downarrow 0$. They defined $\mathsf{LGC} \subset [0, \frac{1}{2}]^{\mathbb{N}}_{\downarrow 0}$ as the family of $p$ for which $\Delta_n \xrightarrow[n \to \infty]{} 0$ and showed that $\mathsf{LGC}$ consists of exactly those $p$ for which

$$T(p) := \sup_{j \in \mathbb{N}} \frac{\log(j+1)}{\log(1/p_j)}, \qquad p \in [0, \tfrac{1}{2}]^{\mathbb{N}}_{\downarrow 0} \quad (2)$$

is finite. They also characterized up to constants the asymptotic decay of $\Delta_n$ (whenever $T(p) < \infty$) via the functional

$$S(p) := \sup_{j \in \mathbb{N}} p_j \log(j+1), \qquad p \in [0, \tfrac{1}{2}]^{\mathbb{N}}_{\downarrow 0}, \quad (3)$$

establishing that $\Delta_n(p)$ decays as $\sqrt{S(p)/n}$. Additional finite-sample bounds provided therein were tightened by Blanchard & Voráček (2024) as follows:

$$\Delta_n(p) \asymp 1 \wedge \left( \sqrt{\frac{S(p)}{n}} + \sup_{j \geq 1} \frac{\log(j+1)}{n \log \left( 2 + \frac{\log(j+1)}{n p_j} \right)} \right),$$

$$\text{if } n \cdot \sup_{j \geq 1} 2 j p_j > 1,$$

$$\Delta_n(p) \asymp \frac{1}{n} \wedge \sum_{j \geq 1} p_j,$$

$$\text{otherwise.}$$

In a later work, Blanchard et al. (2024) extended some of the analysis to the much more difficult case where $\mu$ is not a product measure (i.e., the coordinates of $X \sim \mu$ have correlations). In the present paper, we return to the product-measure case and investigate a different extension: How

[1]Department of Computer Science, Ben-Gurion University of the Negev (BGU), Israel [2]Department of Computer Science, Ariel University, Israel. Correspondence to: Doron Cohen <doronv@post.bgu.ac.il>.

*Proceedings of the 42nd International Conference on Machine Learning*, Vancouver, Canada. PMLR 267, 2025. Copyright 2025 by the author(s).

does $\Delta_n$ behave if rather than restricting the estimator to the MLE $\hat{p}$, we allow *arbitrary* estimators $\tilde{p}$?

Actually, a bit of a refinement in terminology is necessary. When considering (essentially) unrestricted classes of distributions such as those parametrized by $p \in [0, \frac{1}{2}]_{\downarrow 0}^{\mathbb{N}}$, the Empirical Mean Estimator (EME) and the Maximum Likelihood Estimator (MLE) coincide. However, for more general families $\mathcal{P} \subset [0, 1]^{\mathbb{N}}$, this is no longer the case: the likelihood of a given sample might be maximized over $\mathcal{P}$ by some $\hat{p}$ other than the EME. Hence, in the sequel, we shall be pedantic about this distinction, focusing on the EME as the more natural candidate.

Formally, an *estimator* $\tilde{p}$ is any mapping from $(\{0, 1\}^{\mathbb{N}})^n$ to $[0, 1]^{\mathbb{N}}$. Any $p \in [0, 1]^{\mathbb{N}}$ induces the product measure

$$\mu = \mu(p) = \text{Bernoulli}(p_1) \otimes \text{Bernoulli}(p_2) \otimes \ldots \quad (4)$$

on $\{0, 1\}^{\mathbb{N}}$. If $X^{(1)}, X^{(2)}, \ldots, X^{(n)}$ are sampled i.i.d. from $\mu$, then these induce $\tilde{\Delta}_n := \mathbb{E} \|\tilde{p} - p\|_\infty$. We say that a family of product distributions induced by $\mathcal{P} \subset [0, 1]^{\mathbb{N}}$ is *learnable by* $\tilde{p}$ if $\tilde{\Delta}_n \underset{n \to \infty}{\longrightarrow} 0$ for each $p \in \mathcal{P}$, and just *learnable* if it is learnable by some $\tilde{p}$. (Since the sequence $p$ fully determines the measure $\mu(p)$, it is fitting to say that $\tilde{p}$ "learns" $p$ — and hence also $\mu(p)$.)

This general setting immediately raises the natural questions: Can LGC be expanded to a larger learnable family via some estimator $\tilde{p}$ different from the EME? Can some estimator $\tilde{p}$ achieve better decay rates for $\tilde{\Delta}_n$ than the EME?

**Our contributions.** Modulo some technical caveats, we resolve both questions above in the negative. If the learner is barred from exploiting some pathological quirks of the infinite-dimensional setting, then essentially LGC as defined above is the largest learnable family (Theorem 2.1). Furthermore, the EME achieves the minimax risk decay rate over non-pathological distribution families (Theorem 2.2). Finally, in Theorem 2.3 we show that non-trivial extensions of LGC become possible once the restrictions are relaxed.

**Related work.** Estimating the mean of a high-dimensional distribution from independent draws is among the most basic problems of statistics. Much of the earlier theory has focused on obtaining efficient estimators $\hat{m}_n$ of the true mean $m$ and analyzing the decay of $\|\hat{m}_n - m\|_2$ as a function of sample size $n$, dimension $d$, and various moment assumptions on $X$ (Catoni, 2012; Devroye et al., 2016; Lugosi & Mendelson, 2019a;b; Cherapanamjeri et al., 2019; 2020; Diakonikolas et al., 2020; Hopkins, 2020; Lugosi & Mendelson, 2021; Lee & Valiant, 2022). For $d$-dimensional distributions $\mu$ on $\{0, 1\}^d$, Chernoff and union bounds yield $\Delta_n(\mu) \lesssim \sqrt{\ln(d+1)/n}$ for the EME, and a simple information-theoretic argument shows that this is minimax-optimal up to constants (Cohen & Kontorovich,

2023, Proposition 1). Cohen & Kontorovich further motivated their choice of the $\ell_\infty$ norm as the most interesting of all the $\ell_r$ norms, in a well-defined sense (see Blanchard & Voráček (2024, Proposition 7)). Blanchard & Voráček (2024) fully closed the gaps in the analysis of Cohen & Kontorovich, and Blanchard et al. (2024) took the first nontrivial steps in analyzing non-product sampling distributions.

**Notation.** The measure-theoretic subtleties of defining distributions on $\{0, 1\}^{\mathbb{N}}$ are addressed in Cohen & Kontorovich (2023). Our logarithms will always be base e by default; other bases will be explicitly specified. The natural numbers are denoted by $\mathbb{N} = \{1, 2, 3, \ldots\}$ and for $k \in \mathbb{N}$, we write $[k] = \{i \in \mathbb{N} : i \le k\}$. The floor and ceiling functions, $\lfloor t \rfloor$, $\lceil t \rceil$, map $t \in \mathbb{R}$ to its closest integers below and above, respectively; also, $s \vee t := \max\{s, t\}$, $s \wedge t := \min\{s, t\}$. Unspecified constants such as $c, c'$ may change value from line to line. We use superscripts to denote distinct random vectors and subscripts to denote indices within a given vector. Thus, if $X^{(1)}, , \ldots, X^{(n)}$ are independent copies of $X$, then $X_j^{(i)}$ denotes the $j$th entry of the $i$th copy.

When considering the EME as the sole estimator (as in previous works), no generality was lost in restricting the range of $p$ to $[0, \frac{1}{2}]$ and assuming sequences monotonically decreasing to 0 (i.e., $[0, \frac{1}{2}]_{\downarrow 0}^{\mathbb{N}}$). The definitions of $T$ and $S$ in (2, 3) were based on this assumption. In this work, we will need their slightly generalized versions. With the convention $\dot{x} := \min\{x, 1 - x\}$, we define

$$T(p) \quad := \quad \inf_{\sigma: \mathbb{N} \to \mathbb{N}} \sup_{j \in \mathbb{N}} \frac{\log(j+1)}{\log(1/\dot{p}_{\sigma(j)})}, \quad (5)$$

$$S(p) \quad := \quad \inf_{\sigma: \mathbb{N} \to \mathbb{N}} \sup_{j \in \mathbb{N}} \dot{p}_{\sigma(j)} \log(j+1), \quad (6)$$

for $p \in [0, 1]^{\mathbb{N}}$, where the infimum is over all permutations $\sigma$ over $\mathbb{N}$. Whenever $\dot{p}_j \to 0$, a unique non-increasing permutation $\dot{p}^{\downarrow}$ exists, and it is easily seen to be the one achieving both infima above; thus, for $p \in [0, \frac{1}{2}]_{\downarrow 0}^{\mathbb{N}}$, the definitions in (5, 6) coincide with those in (2, 3).

Any $p \in [0, 1]^{\mathbb{N}}$ defines the product measure $\mu = \mu(p)$ as in (4). An *estimator* $\tilde{p}$ and its induced deviation $\tilde{\Delta}_n$ are defined just above (4), and the *learnability* of a family $\mathcal{P} \subset [0, 1]^n$ is defined just below it.

We say that a family $\mathcal{P} \subset [0, 1]^{\mathbb{N}}$ is *decaying* if $\lim_{j \to \infty} \dot{p}_j = 0$ for all $p \in \mathcal{P}$. For $p \in [0, 1]^{\mathbb{N}}$ and $b \in \{-1, 1\}^{\mathbb{N}}$, we say that

$$p' = p'(p, b) \in [0, 1]^{\mathbb{N}}$$

is a *b-reflection* of $p$ about $\frac{1}{2}$ if

$$p'_j = b_j \left( p_j - \frac{1}{2} \right) + \frac{1}{2}, \qquad j \in \mathbb{N}.$$

We say that $\mathcal{P} \subset [0,1]^{\mathbb{N}}$ is *strongly symmetric about* $\frac{1}{2}$ if $p'(p,b) \in \mathcal{P}$ for all $p \in \mathcal{P}$ and $b \in \{-1,1\}^{\mathbb{N}}$.

The family $\mathsf{LGC} \subset [0,\frac{1}{2}]^{\mathbb{N}}$ was defined in Cohen & Kontorovich (2023) as the one learnable by the EME $\hat{p}$, and characterized therein as consisting precisely of those $p \in [0,\frac{1}{2}]^{\mathbb{N}}$ for which $T(p) < \infty$. Since in this work we do not restrict the range of $p$ to $[0,\frac{1}{2}]$, we define

$$\mathsf{L\dot{G}C} \quad := \quad \left\{ p \in [0,1]^{\mathbb{N}} : T(p) < \infty \right\}, \qquad (7)$$

for $T$ as defined in (5). It is straightforward to extend the arguments of Cohen & Kontorovich (2023) to show that $\mathsf{L\dot{G}C}$ consists precisely of those $p \in [0,1]^{\mathbb{N}}$ for which the EME $\hat{p}$ yields $\Delta_n \to 0$.

## 2. Main Results

Our first result may be informally summarized thus: "morally" speaking, $\mathsf{LGC}$ is the largest family that is learnable by any fixed estimator.

**Theorem 2.1** (expanding LGC)**.** *Suppose that* $\mathcal{P} \subset [0,1]^{\mathbb{N}}$ *defines a family of product distributions as in* (4) *and furthermore*

1. $\mathcal{P}$ *is decaying*

2. $\mathcal{P}$ *is strongly symmetric about* $\frac{1}{2}$

3. $\mathcal{P}$ *is learnable.*

*Then* $\mathcal{P} \subseteq \mathsf{L\dot{G}C}$.

**Remark.** Strong symmetry about $\frac{1}{2}$ forces the sequences in $\mathcal{P}$ to be "generic" and prevents the learner from beating the EME by exploiting some special structure. Note that this condition is very much absent in Theorem 2.3, where indeed such exploits become possible.

Having established that (modulo pathologies) $\mathsf{LGC}$ is the largest learnable family, we next show that the EME is nearly minimax-optimal for this family.

**Theorem 2.2** (Minimax bound)**.** *There exist universal constants* $c, c', C > 0$ *such that the following holds. For* $n \in \mathbb{N}$ *and* $s, t > 0$ *satisfying* $\frac{c' \log n}{n} \leq \frac{s}{t} \leq \mathrm{e}^{-1}$, *let*

$$\mathcal{P}_{s,t} := \left\{ p \in [0,1]^{\mathbb{N}} : S(p) \leq s \, \wedge \, T(p) \leq t \right\}.$$

*Then, whenever* $\frac{c' \log n}{n} \leq \frac{s}{t} \leq \mathrm{e}^{-1}$ *and*

$$n \geq \frac{\frac{t^2}{Cs} \log \frac{t}{s}}{\frac{t}{s} \log \frac{t}{s} \cdot \mathrm{e}^{-\frac{t \log \frac{t}{s}}{\log 2}} - 1},$$

*we have*

$$\inf_{\tilde{p}} \sup_{p \in \mathcal{P}_{s,t}} \mathbb{E} \sup_{j \in \mathbb{N}} |\tilde{p}_j - p_j| \geq 1 \wedge \left( c\sqrt{\frac{s}{n}} \, \vee \, Q(t,s) \cdot \frac{t}{n} \right), \tag{8}$$

*where the infimum is over all estimators* $\tilde{p}$ *that are based on* $n$ *i.i.d. samples drawn from* $p$, *and*

$$Q(t,s) = C \left( 1 + \frac{\log \frac{t}{s}}{\log \log \frac{t}{s}} \right)^{-1}.$$

**Remark.** The logarithmic factor and restrictions on the range of $n$ are likely artifacts of the argument, which we kept streamlined for space and readability. We look forward to removing both in the extended version.

Finally, we show that if the learner is allowed to "cheat" by exploiting the information contained in the infinitely many bits of each example $X^{(i)}$, then $\mathsf{LGC}$ can indeed be non-trivially expanded. Let us elaborate a bit on the nature of these exploits. The elements of $\mathsf{LGC}$ have a "generic," unstructured flavor: knowing the values of $p_j$ for $j \in [N]$ reveals no useful information regarding the remaining $j > N$; all the learner knows is that these must decay as some power of $j$ in order to be in $\mathsf{LGC}$. On the other hand, one might consider adjoining a "structured" sequence to $\mathsf{LGC}$, such as $p = (\frac{1}{2}, \frac{1}{2}, \dots)$. Because a single $X^{(i)} \sim \mu$ provides a bit drawn from *each* of the $\mathrm{Bernoulli}(p_j)$, the learner (as we show below) is able to first test whether the unknown sequence has the given structure (in this case, whether it was generated by $p \equiv \frac{1}{2}$) and if not, then reverts to the standard EME for learning the unstructured sequences in $\mathsf{LGC}$.

**Theorem 2.3** (Relaxing decay and symmetry)**.** *Define the family* $\mathrm{const} \subset [0,1]^{\mathbb{N}}$ *by*

$$\mathrm{const} \quad := \quad \{(c,c,\dots) : c \in [0,1]\}.$$

*Then* $\mathcal{P} = \mathsf{L\dot{G}C} \cup \mathrm{const}$ *is learnable, meaning that there exists an estimator* $\tilde{p}$ *such that* $\tilde{\Delta}_n(p) \to 0$ *for all* $p \in \mathcal{P}$.

**Remark.** The techniques of Theorem 2.3 are applicable considerably more broadly than just to the family $\mathcal{Q} = \mathrm{const}$. For example, the argument can be easily adapted to show that $\mathsf{L\dot{G}C} \cup \mathcal{Q}$ is learnable for any finite $\mathcal{Q}$. The following proposition shows a nontrivial $\mathcal{Q}$ which is strongly symmetric and learnable, implying that the decay condition of Theorem 2.1 is necessary.

**Proposition 2.4** (Relaxing decay)**.** *Let* $c > 0$ *be a universal constant, and define the family of distributions*

$$\mathcal{Q} = \left\{ p \in [0,1]^{\mathbb{N}} \mid \forall j \in \mathbb{N}, |p_j - 1/2| \leq \frac{c}{\sqrt{j}} \right\}.$$

*Then,* $\mathcal{Q}$ *is learnable.*

**Open problems.** Two natural directions for future study are extensions of Theorems 2.1 and 2.3. For the former, it is likely that the conditions on $\mathcal{P}$ are too stringent and can be significantly relaxed; in particular, requiring that $\mathcal{P}$ be decaying is quite probably unnecessary. Thus, we seek a larger family $\mathcal{P}'$ whose learnability implies $\mathcal{P}' \subseteq \dot{\mathsf{L}}\mathsf{GC}$. Regarding Theorem 2.3, we again anticipate the existence of considerably richer families $\mathcal{Q}$ for which $\mathsf{LGC} \cup \mathcal{Q}$ is learnable. One such family is proposed in the conjecture below.

**Conjecture.** Let $\mathcal{Q} \subset [0,1]^{\mathbb{N}}$ be a countable family of sequences with the following property: for each $q, q' \in \mathcal{Q}$, there is an $\varepsilon > 0$ and an infinite $J \subset \mathbb{N}$ such that $|q_j - q'_j| > \varepsilon$ for all $j \in J$. Then $\mathsf{LGC} \cup \mathcal{Q}$ is learnable.

## 3. Proofs

### 3.1. Proof of Theorem 2.1

Assume, for the sake of contradiction, that there exists an estimator $\tilde{p}$ and a family $\mathcal{P} \subset [0,1]^{\mathbb{N}}$ satisfying the conditions of the theorem, such that $\mathcal{P}$ is learnable by $\tilde{p}$ but there exists a $p^* \in \mathcal{P} \setminus \dot{\mathsf{L}}\mathsf{GC}$. Based on $p^*$, we will construct a family $\mathcal{P}^* \subset \mathcal{P}$ and argue that $\tilde{\Delta}_n(p) \to 0$ cannot hold for all $p \in \mathcal{P}^*$.

Since $\mathcal{P}$ is strongly symmetric about $\frac{1}{2}$, for any $p \in \mathcal{P}$ and any sign vector $b \in \{-1,1\}^{\mathbb{N}}$, the $b$-reflection $p'(p, b)$ defined by $p'_j = b_j(p_j - \frac{1}{2}) + \frac{1}{2}$ also belongs to $\mathcal{P}$.

Consider the following randomized experiment:

- Let $Y = (Y_j)_{j \in \mathbb{N}}$ be a sequence of independent Rademacher random variables, i.e., $\mathbb{P}(Y_j = 1) = \mathbb{P}(Y_j = -1) = \frac{1}{2}$.

- Define $p^{(Y)} \in [0,1]^{\mathbb{N}}$ as the $Y$-reflection of $p^*$ about $\frac{1}{2}$, i.e. $p_j^{(Y)} = Y_j(p_j^* - \frac{1}{2}) + \frac{1}{2}$.

- Generate $n$ independent draws $X^{(1)}, \ldots, X^{(n)} \in \{0,1\}^{\mathbb{N}}$ from the product distribution $\mu(p^{(Y)})$ as in (4).

The assumption that $\dot{p}_j^* \in [0, \frac{1}{4}]^{\mathbb{N}}$ incurs no loss of generality, since the decay condition implies that $\dot{p}_j^* \leq \frac{1}{4}$ will hold for all sufficiently large $j$. We can ignore $\dot{p}_j \in (\frac{1}{4}, \frac{1}{2}]$ since this would only decrease the estimation error $\tilde{\Delta}_n(p)$.

We follow the standard reduction from the harder problem of estimating $p^{(Y)}$ to the easier problem of recovering the sign vector $y \in \{-1,1\}^{\mathbb{N}}$ that defines the $y$-reflection $p^{(Y)}$. By the Neyman-Pearson lemma, an optimal estimator $\hat{y}$ is one that minimizes the posterior probability of error, i.e.,

$$\hat{y} = \arg\min_{y \in \{-1,1\}^{\mathbb{N}}} \mathbb{P}(Y \neq y \mid X = x),$$

where $Y$ is the random sign vector and $X = \left(X^{(1)}, \ldots, X^{(n)}\right)$ denotes the observed data.

Now

$$
\begin{aligned}
1 - &\mathbb{P}(Y \neq y \mid X = x) \\
&= \mathbb{P}(Y = y \mid X = x) \\
&= \mathbb{P}(Y_1 = y_1 \mid X = x) \cdot \\
&\quad \mathbb{P}(Y_2 = y_2 \mid X = x, Y_1 = y_1) \cdot \\
&\quad \mathbb{P}(Y_3 = y_3 \mid X = x, Y_1 = y_1, Y_2 = y_2) \cdot \ldots
\end{aligned}
$$

Since the $Y_j$ are mutually independent, each of the factors above has the simpler form

$$
\begin{aligned}
\mathbb{P}(Y_k = y_k \mid X = x, Y_1 = y_1, \ldots, Y_{k-1} = y_{k-1}) \\
= \mathbb{P}(Y_k = y_k \mid X = x).
\end{aligned}
$$

We conclude that the events $E_j = \{Y_j \neq y_j \mid X\}$ are mutually independent. Thus

$$
\begin{aligned}
\mathbb{P}(Y \neq y \mid X = x) \\
= \mathbb{P}\left(\bigcup_{j \in \mathbb{N}} E_j\right) \\
= \lim_{N \to \infty} \mathbb{P}\left(\bigcup_{j=1}^{N} E_j\right) \\
= \lim_{N \to \infty} \alpha_N\left(\mathbb{P}(E_1), \mathbb{P}(E_2), \ldots, \mathbb{P}(E_N)\right),
\end{aligned}
$$

where the second equality holds by regularity of probability measures (Kechris, 1995, Theorem 17.10), and $\alpha_N : [0,1]^N \to [0,1]$ is the *inclusion-exclusion* function defined inductively by $\alpha_1(x) = x$ and

$$
\begin{aligned}
\alpha_{N+1}(x_1, x_2, \ldots, x_N, x_{N+1}) \\
= x_{N+1} + (1 - x_{N+1})\alpha_N(x_1, x_2, \ldots, x_N).
\end{aligned}
$$

By Kontorovich (2012, Lemma 4.2), $\alpha_N$ is monotonically increasing in each argument. Hence, the optimal estimator may minimize each $\mathbb{P}(E_j)$ individually — and so we may define $A_j$ as the estimator for the $j$-th coordinate, where $A_j : \{0,1\}^{\mathbb{N} \times n} \to [0,1]$ is any mapping from the $j$-th row of the data matrix to an estimate of $p_j^{(Y)}$. Let $B_j$ be the event that $A_j$ and $p_j^{(Y)}$ belong to different intervals, i.e., either $A_j \in [0, \frac{1}{2})$ and $p_j^{(Y)} \in (\frac{1}{2}, 1]$ or vice versa. To establish a lower bound on the error of any estimator, consider the

minimax risk:

$$\inf_A \sup_{p \in \mathcal{P}} \mathbb{E} \sup_{X} \sup_{j \in \mathbb{N}} |A_j - p_j|$$

$$\geq \inf_A \mathbb{E} \mathbb{E} \sup_{X} \sup_{j \in \mathbb{N}} |A_j - p_j^{(Y)}|$$

$$\geq \inf_A \mathbb{E} \mathbb{E} \sup_{X} \sup_{j \in \mathbb{N}} \mathbf{1}\{B_j\} \left| \frac{1}{2} - p_j^{(Y)} \right|$$

$$\geq \frac{1}{4} \inf_A \mathbb{E} \mathbb{E} \sup_{X} \sup_{j \in \mathbb{N}} \mathbf{1}\{B_j\}$$

$$= \frac{1}{4} \inf_A \mathbb{P}_{Y,X} \left( \bigcup_{j \in \mathbb{N}} B_j \right)$$

$$= \frac{1}{4} \inf_A \int_{x \in \{0,1\}^{\mathbb{N} \times n}} \mathbb{P} \left( \bigcup_{j \in \mathbb{N}} B_j \mid X = x \right) dP_X(x)$$

$$\geq \frac{1}{4} \int_{x \in \{0,1\}^{\mathbb{N} \times n}} \min_{\hat{y} \in \{-1,1\}^{\mathbb{N}}} \mathbb{P}(Y \neq \hat{y} \mid X = x) dP_X(x).$$

By the Neyman-Pearson lemma, the optimal choice of $\hat{y}_j$ is according to the majority vote[1] of the vector $(X_j^{(1)}, \ldots, X_j^{(n)})$. In the event that $(X_j^{(1)}, \ldots, X_j^{(n)}) = (1, 1, \ldots, 1)$, but $1 - \dot{p}_j^* \neq p_j^{(Y)}$, the estimator makes a mistake. The probability of such an event, conditioned on the other random variables $X_{j'}, Y_{j'}$ where $j' \neq j$, is exactly $\frac{1}{2}(\dot{p}_j^*)^n$. Since we assumed $p^* \notin \mathsf{LGC}$ and thus $\dot{p}^* \notin \mathsf{LGC}$, we have $T(\dot{p}^*) = \infty$. Since $\dot{p}_j^* \to 0$, we may assume without loss of generality that it is decreasing monotonically. By Cohen & Kontorovich (2023, Lemma 3), it follows that for all $n \in \mathbb{N}$, we have

$$\sum_{j=1}^{\infty} (\dot{p}_j^*)^n = \infty.$$

Since the events of $\hat{y}_j$ being wrong are mutually independent, the second Borel–Cantelli lemma implies that almost surely at least one of them will occur. It follows that $\liminf_{n \to \infty} \tilde{\Delta}_n(p^{(Y)}) \geq \frac{1}{4}$, contradicting the learnability assumption. $\square$

### 3.2. Proof of Theorem 2.2

We reduce the minimax lower bound problem to one over a finite set of hypotheses. For $2 \leq J \in \mathbb{N}$ and $0 \leq q \leq q' \leq 1/2$ to be chosen below, we consider $J + 1$ profiles $p^{(k)} \in [0, \frac{1}{2}]^{\mathbb{N}}$ for $k \in [J + 1]$. For $k = J + 1$ we take the step profile

$$p_j^{(J+1)} = \begin{cases} q, & j \in [J+1], \\ 0, & j > J + 1, \end{cases}$$

[1]The issue of optimally breaking ties or allowing randomized decision rules is somewhat delicate and is exhaustively addressed in Kontorovich & Pinelis (2019, Eq. (2.7)). In our setting, these do not affect the probability of error.

and for $1 \leq k \leq J$ we take the same step profile but with an additional bump at position $k$,

$$p_j^{(k)} = \begin{cases} q, & j \in [J+1] \text{ and } j \neq k, \\ q', & j = k, \\ 0, & j > J + 1. \end{cases}$$

The construction of these profiles is illustrated in Figure 1.

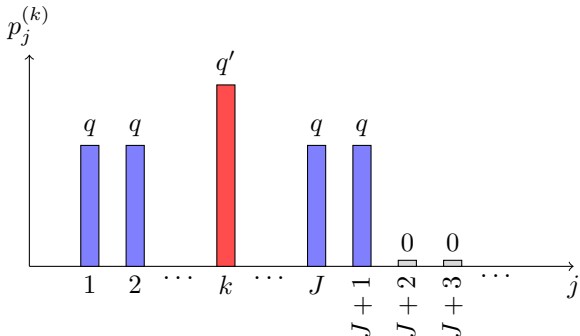

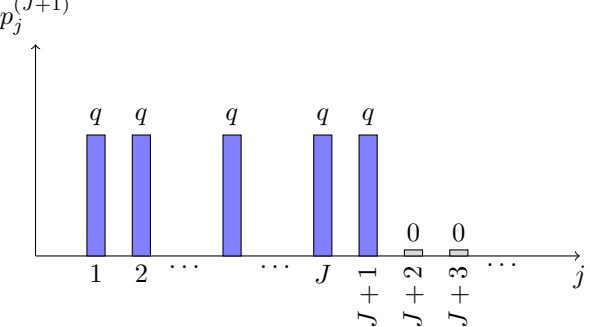

Figure 1. Illustration of the step profile construction for $p^{(k)}$ (top) and the special case for $p^{(J+1)}$ (bottom). Each bar represents the value of $p_j^{(k)}$ at position $j$. Values are shown above the bars.

Note that for all $k \neq \ell \in [J+1]$ we have $\|p^{(k)} - p^{(\ell)}\|_\infty = |q' - q|$. In addition, for $k = J + 1$,

$$S(p^{(J+1)}) = q \log(J + 1)$$

and

$$T(p^{(J+1)}) = \frac{\log(J+1)}{\log \frac{1}{q}},$$

and for $k \in [J]$,

$$S(p^{(k)}) = \max \{q \log(J+1), q' \log 2\}$$

and

$$T(p^{(k)}) = \max \left\{ \frac{\log(J+1)}{\log \frac{1}{q}}, \frac{\log 2}{\log \frac{1}{q'}} \right\}.$$

Given $s \leq \frac{t}{e}$ as in the Theorem statement, we choose $q \in [0, \frac{1}{2}]$ and $J$ as

$$q = \frac{1}{\frac{t}{s} \log \frac{t}{s}} \qquad \text{and} \qquad \log(J + 1) = t \log \frac{t}{s}.$$

Then

$$S(p^{(J+1)}) = q \log(J+1) = s, \tag{9}$$

$$T(p^{(J+1)}) = \frac{\log(J+1)}{\log \frac{1}{q}} = t \cdot \left(1 + \frac{\log \frac{t}{s}}{\log \log \frac{t}{s}}\right)^{-1} \leq t. \tag{10}$$

Below we set $q' \leq 1/2$ such that $q' \geq q$ and for all $k \in [J]$,

$$S(p^{(k)}) \leq S(p^{(J+1)}) = s$$
$$\text{and} \quad T(p^{(k)}) \leq T(p^{(J+1)}) \leq t. \tag{11}$$

Thus, $p^{(k)} \in \mathcal{P}_{s,t}$ for all $k \in [J+1]$ and

$$\inf_{\tilde{p}} \sup_{p \in \mathcal{P}_{s,t}} \mathbb{E} \sup_{j \in \mathbb{N}} |\tilde{p}_j - p_j|$$
$$\geq \inf_{\tilde{p}} \max_{k \in [J+1]} \mathbb{E}_{X_n \sim \mu^{(k,n)}} \|\tilde{p}(X_n) - p^{(k)}\|_\infty. \tag{12}$$

To lower bound the right-hand side of (12) we apply the generalized Fano method. For $k \in [J+1]$, let $\mu^{(k)} = \mu(p^{(k)})$ be the product measure over $\{0,1\}^{\mathbb{N}}$ as defined in (4) and note that $\mathbb{E}_{X \sim \mu^{(k)}}\{X\} = p^{(k)}$. We denote by $\mu^{(k,n)}$ the product measure of $n$ independent copies of $X \sim \mu^{(k)}$. We invoke Lemma 3.1 with the $J+1$ measures $(\nu_1, \ldots, \nu_{J+1}) = (\mu^{(1,n)}, \ldots, \mu^{(J+1,n)})$, the distance function $\rho = \|\cdot\|_\infty$, and the parameters $\theta(\mu^{(k,n)}) = \mathbb{E}_{X \sim \mu^{(k)}}\{X\} = p^{(k)}$ for $k \in [J+1]$. Note that $\rho(\theta(\mu^{(k,n)}), \theta(\mu_\ell^n)) = |q' - q|$ for all $k \neq \ell \in [J+1]$ and that

$$D_{\text{KL}}(\mu^{(k,n)} \| \mu_\ell^n) \leq n(h(q\|q') + h(q'\|q)),$$

where

$$h(q\|q') = q \log \frac{q}{q'} + (1-q) \log \frac{1-q}{1-q'}.$$

Then Lemma 3.1 implies

$$\inf_{\tilde{p}} \max_{k \in [J+1]} \mathbb{E}_{X_n \sim \mu^{(k,n)}} \|\tilde{p}(X_n) - p^{(k)}\|_\infty$$
$$\geq \frac{q' - q}{2} \left(1 - \left(\frac{n(h(q\|q') + h(q'\|q)) + \log 2}{\log(J+1)}\right)\right). \tag{13}$$

We now fix $q'(q) = q'(q, n, J) \geq q$ to be the solution to the equation

$$h(q\|q'(q)) + h(q'(q)\|q) = \frac{\log(J+1)}{2Cn}. \tag{14}$$

Below we verify that (11) indeed holds with this choice of $q'(q)$. Substituting (14) into (13), we obtain the lower bound

$$\frac{q'(q) - q}{2} \left(1 - \left(\frac{\log(J+1)/C + \log 2)}{\log(J+1)}\right)\right)$$
$$\geq \frac{q'(q) - q}{2} \left(1 - \frac{1}{C} - \frac{\log 2}{\log 3}\right)$$
$$\geq \frac{q'(q) - q}{8},$$

for an appropriate value of the constant $C > 0$.

We analyze $q'(q) - q$ for $q'(q)$ satisfying (14) as in Blanchard et al. (2024) and consider two regimes for $h(q\|q') + h(q'\|q)$. For any $0 < q \leq q' \leq \frac{1}{2}$, we have

$$\frac{(q' - q)^2}{q'} \leq h(q\|q') + h(q'\|q) \leq \frac{2(q' - q)^2}{q}. \tag{15}$$

So, by the right inequality in (15),

$$q'(q) - q \geq \sqrt{\frac{q(h(q\|q'(q)) + h(q'(q)\|q))}{2}}$$
$$= \sqrt{\frac{q \log(J+1)}{4Cn}}$$
$$= \sqrt{\frac{S(p_{J+1})}{4Cn}}$$
$$= \sqrt{\frac{s}{4Cn}}. \tag{16}$$

In addition, by the left inequality in (15),

$$q' \leq q + \sqrt{\frac{q' \log(J+1)}{2Cn}}$$
$$\leq \sqrt{q'} \left(\sqrt{q} + \sqrt{\frac{\log(J+1)}{2Cn}}\right),$$

which implies

$$q' \leq \left(\sqrt{q} + \sqrt{\frac{\log(J+1)}{2Cn}}\right)^2$$
$$= q + 2\sqrt{\frac{q \log(J+1)}{2Cn}} + \frac{\log(J+1)}{2Cn}$$
$$= q \left(1 + 2\sqrt{\frac{\log(J+1)}{2Cnq}} + \frac{\log(J+1)}{2Cnq}\right).$$

Since by assumption $Cqn = \frac{Cn}{\frac{t}{s} \log(\frac{t}{s})} \geq c' \log 2$, we have that for a sufficiently large constant $c'$,

$$q' \leq \frac{q \log(J+1)}{\log 2}.$$

This verifies (11) and establishes the term $c\sqrt{\frac{s}{n}}$ in (8).

Next, we assume $\frac{t}{n} \geq c\sqrt{\frac{s}{n}}$. For any $0 \leq q \leq q' \leq 1/2$ we have $h(q\|q') \leq h(q'\|q)$, and for $q' \geq 9q$, it holds that (see, e.g., Blanchard et al. (2024))

$$\frac{1}{2} \leq \frac{h(q\|q') + h(q'\|q)}{q \cdot \frac{q' - q}{q} \log \frac{q' - q}{q}} \leq 4. \tag{17}$$

For $z \geq e$, the solution $x$ to the equation $x \log x = z$ satisfies $x \geq \frac{z}{\log z}$ (Corless et al., 1996). Taking $c > 0$ sufficiently large such that

$$z = \frac{\log(J+1)}{2Cqn} = \frac{t^2}{2Cns} \log^2\left(\frac{t}{s}\right) \geq \frac{c^2}{2C} \geq e,$$

we have that $q'(q)$ satisfies (14) if $q'(q) - q \geq 8q$ and

$$\frac{q'(q) - q}{q} \geq \frac{\frac{\log(J+1)}{2Cqn}}{\log \frac{\log(J+1)}{2Cqn}};$$

namely,

$$
\begin{aligned}
q'(q) - q &\geq \frac{\log(J+1)}{2Cn \log\left(\frac{\log(J+1)}{2Cqn}\right)} \\
&= \frac{T(p_{J+1})}{2Cn} \cdot \frac{\log \frac{1}{q}}{\log \frac{\log(J+1)}{2Cqn}} \\
&= \frac{T(p_{J+1})}{2Cn} \cdot \frac{\log \frac{1}{q}}{\log \frac{s}{2Cnq^2}} \\
&\geq \frac{T(p_{J+1})}{4Cn} \\
&= \frac{t}{4Cn} \cdot \left(1 + \frac{\log \frac{t}{s}}{\log\log \frac{t}{s}}\right)^{-1},
\end{aligned}
$$

where in the last inequality we used the fact that $\frac{s}{2Cn} \leq 1$. Lastly, we verify that $q'(q)$ is such that (11) holds, namely, $\frac{\log \frac{1}{q}}{\log \frac{1}{q'(q)}} \leq \frac{\log(J+1)}{\log 2}$. The left inequality in (17) implies that $q'(q) \leq q + C' \frac{t}{n}$ for some constant $C'$. Putting this and $q = \frac{1}{\frac{t}{s} \log \frac{t}{s}}$ and $\log(J+1) = t \log \frac{t}{s}$, we have that (11) holds if

$$\frac{\log(\frac{t}{s} \log \frac{t}{s})}{\log(\frac{t}{s} \log \frac{t}{s}) - \log(1 + \frac{t^2}{Cns} \log \frac{t}{s})} \leq \frac{t \log \frac{t}{s}}{\log 2}.$$

This is satisfied when

$$n \geq \frac{\frac{t^2}{Cs} \log \frac{t}{s}}{\frac{t}{s} \log \frac{t}{s} \cdot e^{-\frac{t \log \frac{t}{s}}{\log 2}} - 1}.$$

Finally, we consider the case where $t \geq n$. We repeat the arguments in the proof of Theorem 2.1 to show that in this case the minimax rate is bounded from below by a constant. Taking any $p^* \in \mathcal{P}_{s,t}$ such that $T(p^*) \geq n$, and assuming without loss of generality that $p^*$ is non-increasing, let $j'$ be such that

$$T(p^*) \geq \frac{\log(1 + j')}{\log(1/\dot{p}^*_{j'})}.$$

Then

$$
\begin{aligned}
\sum_{j=1}^{\infty} (\dot{p}^*_j)^n &\geq \sum_{j=1}^{\infty} (\dot{p}^*_j)^{T(p)} \\
&\geq \sum_{j=1}^{j'} (\dot{p}^*_j)^{\frac{\log(1+j')}{\log(1/(\dot{p}^*_j))}} \\
&\geq j'(\dot{p}^*_{j'})^{\frac{\log(1+j')}{\log(1/(\dot{p}^*_{j'}))}} \\
&= \frac{j'}{1 + j'} \geq \frac{1}{2}.
\end{aligned}
$$

As in the proof of Theorem 2.1, applying Lemma 3.2 with

$A_j =$
$$\left\{(X_j^{(1)}, \ldots, X_j^{(n)}) = (1, 1, \ldots, 1), \text{ but } 1 - \dot{p}^*_j \neq p_j^{(Y)}\right\},$$

where $Y_j \sim \text{Bernoulli}(1/2)$ and $p_j^{(Y_j)} = Y_j \dot{p}^*_j + (1 - Y_j)(1 - \dot{p}^*_j)$, we get that the minimax rate is lower bounded by a universal constant. □

### 3.3. Proof of Theorem 2.3

We aim to prove that the family

$$\mathcal{P} := \mathsf{L\dot{G}C} \cup \{(c, c, \ldots) : c \in [0, 1]\}$$

is learnable by an estimator $\tilde{p}_n$. Choose some $p \in \mathcal{P}$. The general strategy is to construct an estimator that can distinguish between cases where $T(p) = \infty$ and cases where $T(p) < \infty$, based on the sample.

**Step 1: Testing if $T(p) = \infty$.** We begin by defining a test $\Phi$ to check whether $T(p) = \infty$. The idea is to check the first half of the sequence $X_j^{(1)}, X_j^{(2)}, \ldots, X_j^{(n)}$ are all ones and the second half are all zeros. Formally, the test function is defined as:

$$\Phi(X^{(1)}, X^{(2)}, \ldots, X^{(n)}) = \mathbf{1}\left(\limsup_{j \to \infty} E_j\right),$$

where we define the event

$$E_j := \left\{X_j^{(i)} = 0 \text{ for } i \leq \frac{n}{2} \text{ and } X_j^{(i)} = 1 \text{ for } i > \frac{n}{2}\right\}.$$

Note that, for each $j$, we have

$$\mathbb{P}(E_j) = p_j^{\lfloor n/2 \rfloor}(1 - p_j)^{\lceil n/2 \rceil},$$

and because $\{E_j\}_j$ are independent, by the two Borell-Cantelli lemmas, $\Phi = 1$ almost surely if and only if

$$\sum_{j=1}^{\infty} p_j^{\lfloor n/2 \rfloor}(1 - p_j)^{\lceil n/2 \rceil} = \infty.$$

The above sum can be estimated by the following sums,

$$\sum_{j=1}^{\infty} \dot{p}_j^n \leq \sum_{j=1}^{\infty} p_j^{\lfloor n/2 \rfloor}(1 - p_j)^{\lceil n/2 \rceil} \leq \sum_{j=1}^{\infty} \dot{p}_j^{\lfloor n/2 \rfloor}. \quad (18)$$

**Step 2: Consistency of the Test.** We now show that the test $\Phi$ is consistent. First, assume $T(p) = \infty$, which means $T(\dot{p}_{\downarrow 0}) = \infty$, then by Cohen & Kontorovich (2023, Lemma 3), we have $\sum_{j=1}^{\infty} \dot{p}_j^n = \infty$ for all $n$, then by (18) we have $\Phi = 1$ almost surely.

On the other hand, if $T(p) < \infty$, again by Cohen & Kontorovich (2023, Lemma 3), we have $\sum_{j=1}^{\infty} \dot{p}_j^n < \infty$ for large enough $n$, which means that for large enough $n$ we have $\Phi = 0$ almost surely, as before.

**Step 3: Defining the Estimator.** Once the test $\Phi$ has been applied, we define the estimator $\tilde{p}_n$ as follows:

$$\tilde{p}_n(j) = \begin{cases} \frac{1}{n} \sum_{i=1}^{n} \hat{p}_n(i), & \text{if } \Phi(X_1, \ldots, X_n) = 1, \\ \hat{p}_n(j), & \text{otherwise.} \end{cases}$$

In words, if the test $\Phi$ indicates that $T(p) = \infty$, we use the average of all $\hat{p}_n(j)$ (as this is consistent with the assumption that $p$ is a constant sequence). If $\Phi$ indicates that $T(p) < \infty$, we use the EME $\hat{p}_n(j)$ directly.

**Step 4: Consistency of the Estimator.** We now verify that the estimator $\tilde{p}_n$ is consistent.

If $p \in \mathsf{LGC}$, then $T(p) < \infty$, and the test $\Phi$ will eventually return 0. In this case, the estimator $\tilde{p}_n(j)$ is simply the EME, which is known to be consistent for all $p \in \mathsf{LGC}$. Therefore, $\tilde{p}_n \to p$ in $\ell_\infty$ as $n \to \infty$.

If $p = (c, c, \ldots)$ for some constant $c \in [0, 1]$, then $T(p) = \infty$, and the test $\Phi$ will always return 1. In this case, the estimator $\tilde{p}_n(j)$ is the average of all $\hat{p}_n(j)$, which, by the law of large numbers, will converge to $c$. Thus, $\tilde{p}_n(j) \to c$ as $n \to \infty$.

**Conclusion.** The estimator $\tilde{p}_n$ correctly learns all distributions in the family $\mathsf{L\dot{G}C} \cup \{(c, c, \ldots) : c \in [0, 1]\}$, completing the proof.

$\square$

### 3.4. Proof of Proposition 2.4

Define the estimator $\tilde{p}_n(j)$ as follows:

- For indices $j \leq k(n)$, where $k(n)$ is chosen later, estimate $p_j$ using the empirical mean:

$$\tilde{p}_n(j) = \frac{1}{n} \sum_{i=1}^{n} X_j^{(i)},$$

  where $X_j^{(i)} \sim \text{Bernoulli}(p_j)$.

- For indices $j > k(n)$, set $\tilde{p}_n(j) = 1/2$.

Using Chernoff's bound, for any $\epsilon > 0$,

$$\mathbb{P}\left(|\tilde{p}_n(j) - p_j| \geq \epsilon\right) \leq 2\exp(-2n\epsilon^2).$$

Applying the union bound over $j \leq k(n)$,

$$\mathbb{P}\left(\sup_{j \leq k(n)} |\tilde{p}_n(j) - p_j| \geq \epsilon\right) \leq 2k(n)\exp(-2n\epsilon^2).$$

Choosing

$$\epsilon_n = O\left(\sqrt{\frac{\log k(n)}{n}}\right),$$

ensures this probability vanishes. Thus, with high probability,

$$\sup_{j \leq k(n)} |\tilde{p}_n(j) - p_j| = O\left(\sqrt{\frac{\log k(n)}{n}}\right).$$

For $j > k(n)$, since $p_j$ is approximated by $1/2$:

$$\sup_{j > k(n)} |p_j - 1/2| \leq \sup_{j > k(n)} \frac{c}{\sqrt{j}} = O\left(\frac{1}{\sqrt{k(n)}}\right).$$

Choosing $k(n) = \Theta(n)$ ensures this term vanishes.

Thus, combining both terms,

$$\mathbb{E}\|\tilde{p}_n(j) - p\|_\infty = O\left(\sqrt{\frac{\log k(n)}{n}}\right) + O\left(\frac{1}{\sqrt{k(n)}}\right),$$

which converges to zero as $n \to \infty$. Therefore, $\mathcal{Q}$ is learnable.

$\square$

### 3.5. Auxiliary lemmas

**Lemma 3.1** (Yu (1997)). *For $r \geq 2$, let $\nu_1, \nu_2, \ldots, \nu_r$ be a collection of $r$ probability measures with some parameter of interest $\theta(\nu)$ taking values in pseudo-metric space $(\Theta, \rho)$ such that for all $j \neq k$,*

$$\rho(\theta(\nu_j), \theta(\nu_k)) \geq \alpha$$

*and*

$$D_{\mathrm{KL}}(\nu_j \| \nu_k) \leq \beta.$$

*Then*

$$\inf_{\hat{\theta}} \max_{k \in [r]} \mathbb{E}_{Z \sim \nu_k} \rho(\hat{\theta}(Z), \theta(\nu_k)) \geq \frac{\alpha}{2}\left(1 - \left(\frac{\beta + \log 2}{\log r}\right)\right),$$

*where the infimum is over all estimators $\hat{\theta} : Z \mapsto \Theta$.*

**Lemma 3.2** (Van Handel (2014) Problem 5.1a). *If $A_1, \ldots, A_N$ are independent events, then*

$$(1 - \mathrm{e}^{-1})\left[1 \wedge \sum_{k=1}^{N} \mathbb{P}(A_k)\right] \leq \mathbb{P}\left(\bigcup_{k=1}^{N} A_k\right).$$

## Acknowledgements

The authors gratefully acknowledge financial support from the *Data Science Research Center* at Ben-Gurion University of the Negev (DSRC–BGU) through its 2025 Student Travel Support Program, which funded their participation in ICML 2025. This support is deeply appreciated. Roi Weiss was supported in part by the Israel Science Foundation (grant No. 2442/24)

## Impact Statement

This paper presents work whose goal is to advance the field of Machine Learning. There are many potential societal consequences of our work, none which we feel must be specifically highlighted here.

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

# A. Simulation Results

To support our theoretical findings, we present two sets of simulations. The first demonstrates the tightness of the lower bound in Theorem 2.2, while the second highlights a specific setting where the simple average estimator outperforms the Empirical Mean Estimator (EME), complementing the results of Theorem 2.3.

## A.1. Tightness of the Lower Bound in Theorem 2.2

The first simulation aims to validate the theoretical bounds presented in Theorem 2.2. Specifically, we compare the empirical average supremum deviation $\Delta_n$ with the theoretical predictions for different values of $q$ (variance control parameter) and sample sizes $n$.

We consider six values of $q$: $q = 0.1$, $q = 0.2$, $q = 0.05$, $q = 0.01$, $q = 0.005$, and $q = 0.002$. For each configuration, empirical results are averaged over $J = 100$, 1000, and 10000 repetitions to ensure stability. The empirical deviations are plotted alongside theoretical predictions in a log-log scale to capture the decay behavior as $n$ increases.

Figure 2 shows the results. The empirical deviations (dashed lines) closely follow the theoretical bounds (solid lines), confirming the tightness of the lower bound in Theorem 2.2. As expected, larger values of $J$ lead to smoother empirical curves, emphasizing the role of averaging in reducing variance. Notably, the empirical deviations converge to the theoretical decay rate as $n$ grows.

## A.2. Performance Comparison: Theorem 2.3 Complement

In the second simulation, we explore a specific setting where the simple average estimator surpasses the performance of the EME. This complements the findings of Theorem 2.3 by demonstrating that allowing certain structured distributions can yield better decay rates with alternative estimators.

We evaluate the performance of the EME and the simple average estimator under six different distributions: uniform, triangular, Beta(2,2), exponential, $1/n$-scaled, and Gaussian. For each distribution, we vary the number of trials $k \in \{10, 50, 100, 500\}$ and compute the error as a function of the sample size $n$. The results, plotted on a log-log scale, are shown in Figure 3.

The plots reveal that the simple average estimator achieves lower error rates compared to the EME across all settings as $k$ increases. This improvement is most pronounced for structured distributions like $1/n$-scaled, Beta(2,2), and Gaussian, where the averaging process effectively captures the underlying structure. These findings corroborate the theoretical insights of Theorem 2.3, showcasing that the choice of estimator can significantly impact performance in specific scenarios.

## A.3. Discussion

The results of these simulations provide strong empirical support for our theoretical findings. The first simulation confirms the tightness of the lower bound in Theorem 2.2, while the second demonstrates the practical advantages of alternative estimators, as predicted by Theorem 2.3. These findings highlight the robustness and relevance of our theoretical framework for analyzing the Local Glivenko-Cantelli (LGC) class.

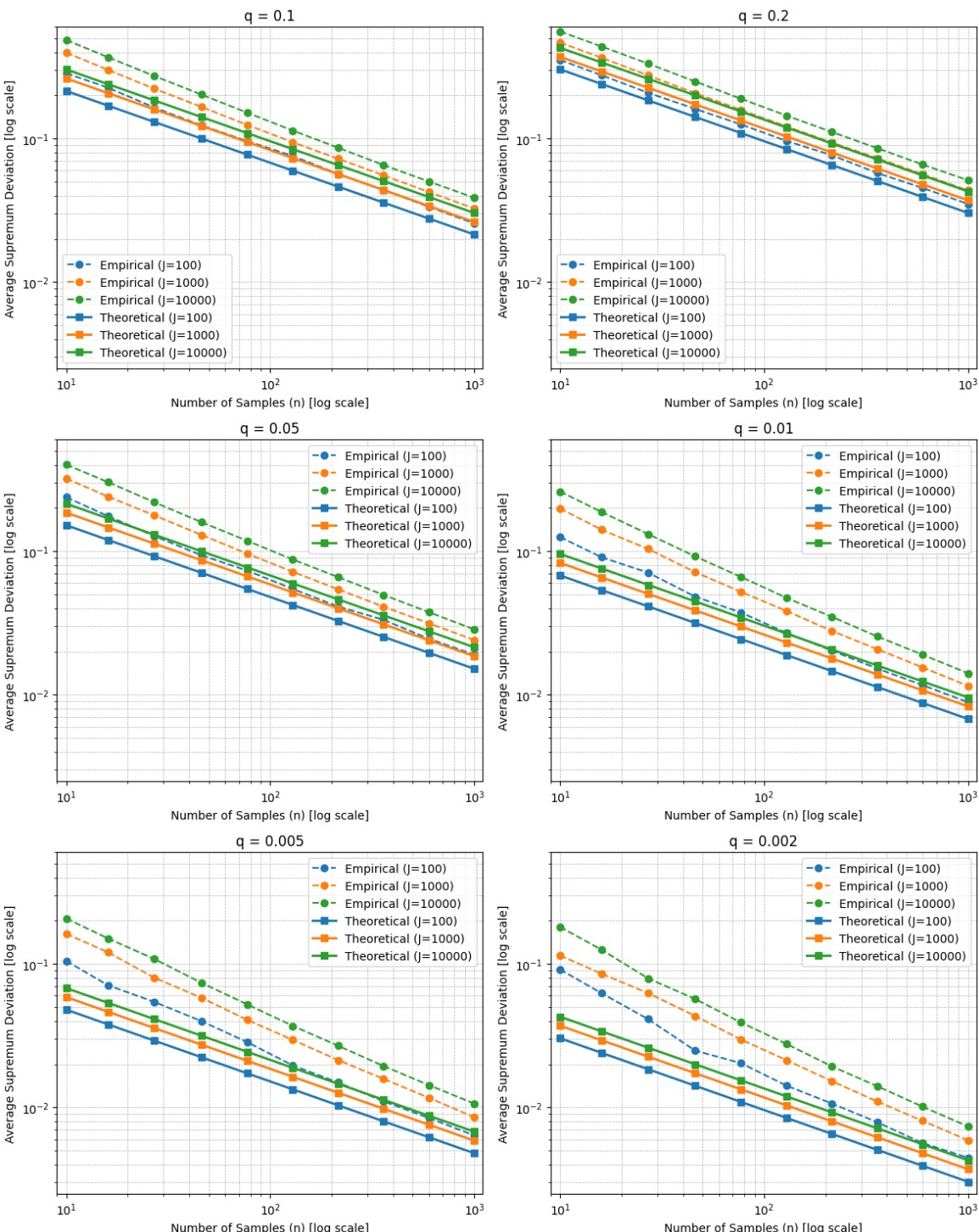

*Figure 2.* Average supremum deviation $\Delta_n$ as a function of sample size $n$ on a log-log scale for varying $q$ values ($q = 0.1, 0.2, 0.05, 0.01, 0.005, 0.002$). Empirical results (dashed lines) are averaged over $J = 100$, 1000, and 10000 repetitions and are compared to theoretical predictions (solid lines).

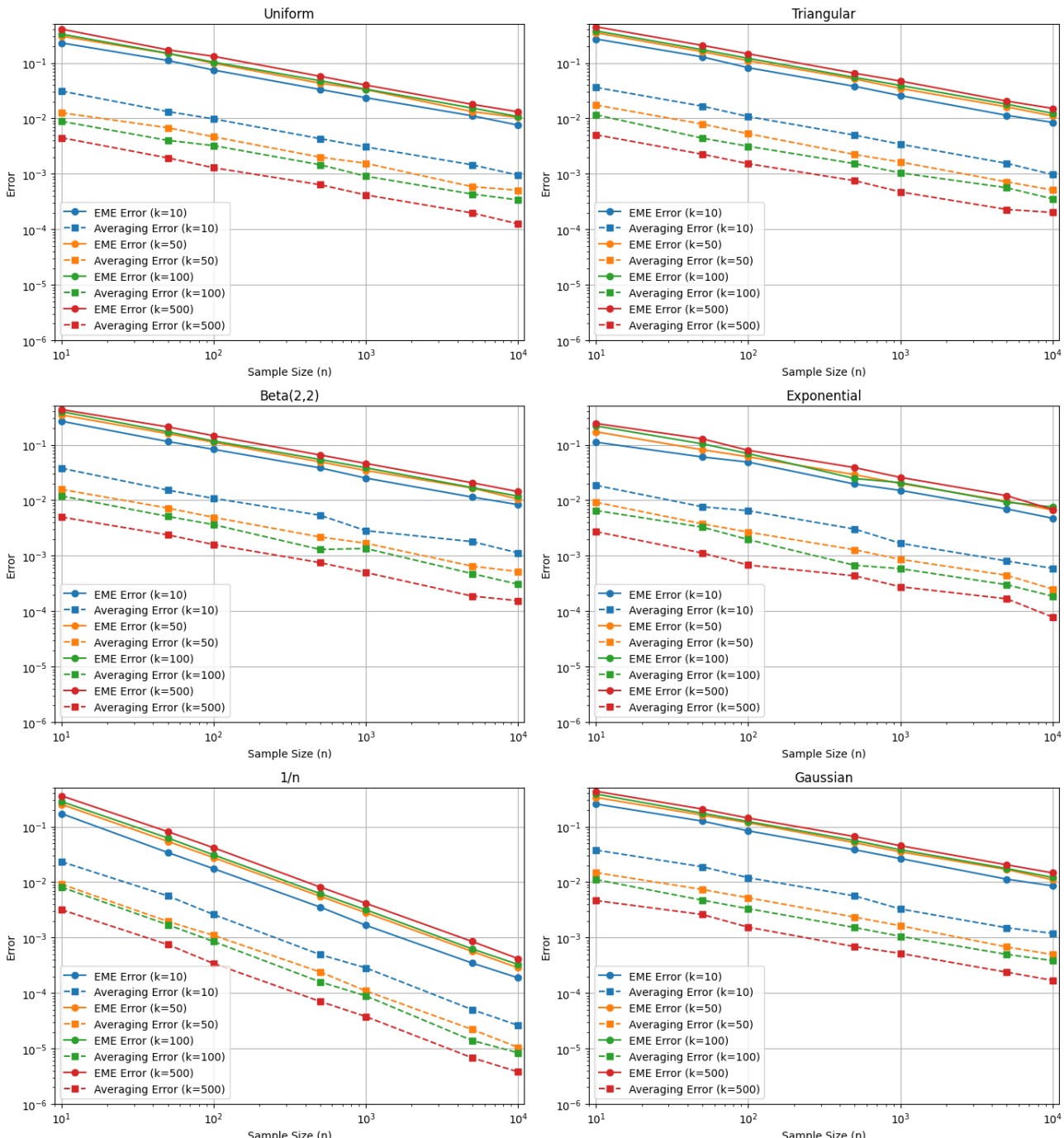

*Figure 3.* Error comparison between the EME and the simple average estimator for varying sample sizes $n$ under different distributions: uniform, triangular, Beta(2,2), exponential, $1/n$, and Gaussian. Results are plotted for $k \in \{10, 50, 100, 500\}$ to illustrate the effect of averaging.

