# OpenReview forum: "The Empirical Mean is Minimax Optimal for Local Glivenko-Cantelli"
_ICML.cc/2025/Conference — ICML 2025 poster_

### Official Review · Reviewer_vocu · 2025-02-27

**Overall Recommendation:** 3

**Summary:**

Background: In this work the authors investigate the question of estimating densities on $\lbrace 0,1\rbrace^\mathbb{N}$ where it is assumed that each index is independent and a sample has the form of a sequence of iid $Bern(p_i)$ random variables, so the density has the form $\mu = \prod_{i \in \mathbb{N}} \mu_i$, from iid samples of $\mu$. The authors are interested in the estimation loss of $\Delta_ n= \mathbb{E}\Vert p - \hat{p}\Vert_\infty$. Its not hard to see that without any assumptions on $(p_i)^\infty_{i=1}$, one cannot control $\Delta_n$ for any estimator, and especially not using the empirical PMF (EME). Previous work has looked into this and established a class of sequences $\mathsf{LGC}$ where the $\Delta_n$ can be bounded for the EME. In particular sequences $\mathsf{LGC}$ are sequences of $[0,1]^\mathbb{N}$ where all entries are less than or equal to 1/2, $p_i \downarrow 0$ and $(p_i)_{i=1}$ must go to zero a certain rate ((2) in the paper).

Contribution: In this work the authors look at relaxed versions of the above setting by considering estimators other than EME and larger classes of densities. The larger class of densities they look at a class that is similar to $\mathsf{LGC}$, which we will call $\mathcal{P}$. In this $\mathcal{P}$ the authors relax $p \in [0,1/2]^\mathbb{N}$ to $p \in [0,1]^\mathbb{N}$ and instead of sequences going to zero they must concentrate near $\lbrace 0,1 \rbrace$ as $i\to nfty$: so $1/2- |p_i-1/2| \infty$, and that, for all $p \in \mathcal{P}$ fliping any number of entires (finite or infinite) about 1/2 (e.g. 0.1 \to 0.9) also lies in $p$. In Thm 2.1 the authors show that any learnable set of $\mathcal{P}$ must have a decay of $1/2- |p_i-1/2| \infty$ at the same rate as $\mathsf{LGC}$ thereby tightly delineating the boundary of this expanded collection of densities and showing that this expanded $\mathsf{LGC}$ in some sense contains all learnable densities. In Thm 2.2 they show that EME is nearly minimax optimal for learning $\mathsf{LGC}$. I would call these the _core_ results of the work, with 2.3 and 2.4 exploring the boundaries of the setting a bit more.

## Update after rebuttal

I have raised my score by one. While I appreciate the clarifications provided in the rebuttal, I still find the contribution to be relatively modest for a purely theoretical result. It is uncommon for a theory paper to present a complete proof within the main text, which suggests a limited depth. Although the significance remains somewhat unclear in my view, the rebuttal has helped to improve my assessment in that regard. I would strongly recommend that the authors include more context regarding the practical implications (or potential practical implications) of their work.

**Claims And Evidence:**

N/A, the work is purely theoretical.

**Essential References Not Discussed:**

This seemed fine to me.

**Experimental Designs Or Analyses:**

N/A

**Methods And Evaluation Criteria:**

N/A, the work is purely theoretical.

**Other Comments Or Suggestions:**

- l. 44 left: Fix aligment with $\mathbb{E} X^{(1)}$.
- l. 121 left: "morally" sounds very strange to me here.
-l. 144 right vs 158 right: I think I'm just missing an implication of your results but it see "implying that the decay condition of Theorems 2.1 is necessary." and "Two natural directions for future study are extensions of Theorems 2.1 and 2.3. For the former, it is likely that the conditions on P are too stringent and can be significantly relaxed; in particular, requiring that P be decaying is quite probably unnecessary." Which seem to contradict each other. Maybe this should be elaborated on a bit.

I don’t see any major issues with the work itself, but the results seem like such a poor fit for ICML that I was considering giving it a 1 (Reject). As someone who regularly submits highly theoretical papers to major ML conferences, I would expect this work to receive virtually no attention in this venue. It would likely have much more impact at COLT or another more topically appropriate venue, such as ITA.

**Other Strengths And Weaknesses:**

There are a few issues with this work:

- While I personally appreciate fundamental questions about learnability, I don’t think this topic is a good fit for ICML. This paper extends work from a previous COLT paper, which seems like a more appropriate venue. The broader significance for the ML community is quite limited—I struggle to see any practical applications or significant implications beyond a purely academic exploration of density estimation. Theory papers at ICML should either offer insights relevant to practitioners or shed light on important settings, and this work does neither.

- The proofs are fairly standard. If the paper introduced a genuinely novel technique or a powerful new idea, it might justify acceptance despite its lack of practical impact. However, everything here looks quite routine to me.

**Questions For Authors:**

My main question is: why should the broader ML community be interested in these results? Does this type of estimation problem arise in any important setting? From the name, I assume there’s some connection to CDF estimation, but the link isn’t clear to me. I’m actually surprised by how little motivation is provided, especially for an ICML paper.

**Relation To Broader Scientific Literature:**

This work is an extension of a line of research that was started very recently on estimating densities on infinite bit strings, where all the bits are independent. Judging from the recent research this is, surprisingly, a setting that has only been explored recently. Beyond this it is difficult for me to contextualize this more. This work seems to be a natural extension of discrete density estimation to an alphabet that is uncountable. The rates achieved match the well known $1/\sqrt{n}$ rate for estimation over a finite or countable alphabet. Naturally we need some regularity conditions (functions T and S) to make this problem tractable. This work relaxes the regularity condition and shows that a previous work was minimax optimal, which is always nice to know.

**Theoretical Claims:**

I checked the proofs Theorems 2.1 and 2.2 somewhat carefully and did not find any significant issues although I admittedly did not do a 100% assiduous check of these proofs, e.g., check the references or double check every single step of algebra. The proof techniques are very standard, in a setting that isn't terribly delicate, and the general techniques and results align well with what one would expect in this setting. That being said, I think there may be a few (non-critical) issues.

* I think there may be a small error on l. 188 right. Should "$\alpha_1(x) = 1$" instead be "$\alpha_1(x) = x$"? For clarity I would just write $\alpha_1(x_1) = x_1$. My understanding is that we want to write $P(\cup E_i)$ as the $P(E_1) + P(E_1^C\cap E_2) + P(E_1^C \cap E_2^C \cap E_3) + \cdots$, where we replace the intersects with products due to independence, in which case we want $\alpha_1(P(E_1)) = P(E_1)$.

* Line 197 left: I agree that this assumption should be nonproblematic, but I think this should be dealt with rigorously.

* $\mu^{(k,n)}$ is defined three sentences later. Please introduce this before or immediately after using it.

---

> ### Author Rebuttal · Authors · 2025-03-31
>
> Thank you for your thorough evaluation. We address your concerns below:
>
> 1. Relevance and Venue Suitability
>    We recognize that our work is theoretical. However, uniform convergence in the Local Glivenko-Cantelli sense is fundamental to learning theory, informing both risk bounds and distribution‐estimation methods. While we considered submitting to a more theory‐focused venue such as COLT, we also believe there is an audience at ICML that values rigorous foundational work in ML.  We can point to dozens of papers published in ICML at a comparable level of theoretical abstraction.
>
> 2. Minor Theoretical / Textual Points
>    - Line 188 (right): We agree that $\alpha_1(x)$ should be $\alpha_1(x)=x$.
>    - Line 197 (left): We will justify the assumption more rigorously.
>    - Definition Timing: We will introduce $\mu^{(k,n)}$ earlier for clarity.
>    - Line 44 (left): We will correct the alignment with $\mathbb{E}X^{(1)}$.
>    - Line 121 (left): We will replace “morally” with a more precise term.
>    - Lines 144 (right) vs. 158 (right): We acknowledge the apparent contradiction concerning the necessity versus the stringency of the decay condition. We do believe the condition can be replaced with weaker assumptions in certain scenarios, and we will clarify this nuance.
>
> 3. Broader Motivation and Applications
>    Although the problem may appear standalone, it initially arose from analyzing practical algorithms, as detailed in the Local Glivenko-Cantelli paper by Cohen & Kontorovich (2023), which stems from the multiplicative weights method. Additional ties to CDF estimation are found in the work of Blanchard & Voráček (2024), where a modified Dvoretzky-Kiefer-Wolfowitz (DKW) approach is employed. Given the inherent space limitations of a conference paper, we chose to prioritize presenting novel results over a recapitulation of well-established motivation in recent literature. That said, we take this critique to heart and will expand upon the motivation -- including genuinely "applied" ramifications -- in the revision. In particular, we will delve into the original motivation behind the LGC question: Multiplicative Weights algorithm for online learning or solving a repeated zero-sum game, with a distribution-dependent, dimension-free dependence on the number of "experts".
>
> We appreciate your feedback and hope these clarifications address your concerns while underscoring both the importance and rigor of our contributions.

---

> > ### Comment · Reviewer_vocu · 2025-04-04
> >
> > Thank you for you rebuttal.
> >
> > I am inclined inclined to maintain my score. I agree that this is a reasonably established line of work, but I don't feel comfortable asserting that this is a significant result in the context of ICML. Looking at the  "Local Glivenko-Cantelli paper by Cohen & Kontorovich (2023)" the significance is still not clear to me (the paper contains no mention of the multiplicative weights method).
> >
> > Again I am still somewhat on the fence about this, but it still seems as though the authors are not capable of concretely and clearly describing why the findings in the paper are significant to the general ML community: e.g., clearly describing the learning problem and method for which this gives some interesting or useful implications. Looking at the other reviews it seems like the other reviewers don't really know why these results are significant either, and in light of that I lean towards reject.

---

> > > ### Author Response · Authors · 2025-04-04
> > >
> > > Clearly, for the ML community, the importance of high-dimensional mean estimation is undisputed. Any method that replaces the worst-case dependence on the dimension by a much more refined dependence on the distribution can considerably sharpen the bounds. Consider, for example, the original problem statement motivating the line of results on LGC, originating in this cstheory post:
> > >
> > > https://cstheory.stackexchange.com/questions/42009/is-uniform-convergence-faster-for-low-entropy-distributions
> > >
> > > In systems with many experts, bounds with worse-case dependence on the dimension (i.e., the number of experts) are useless when we sample from experts whose expertise adapts dynamically (the dynamics of the multiplicative weight updates mentioned in the post is one such example). Our results show that when the algorithm progresses and the distribution becomes less entropic, less samples are needed to ensure uniform estimation of the experts' expertise, and estimating with the empirical mean is optimal.
> > >
> > > We fully intend to pursue this and other applications in an active current line of research. The thrust of the present results is to examine the optimality of the empirical mean estimator for this problem. Insofar as the original motivation is compelling, we argue that understanding the optimal estimator for it is equally important.

---

### Official Review · Reviewer_4ufE · 2025-03-08

**Overall Recommendation:** 4

**Summary:**

In the local Glivenko-Cantelli setting, one seeks to learn an unknown distribution $\mu$ over $\{0,1\}^\mathbb{N}$ from samples. In the case where $\mu$ is a product measure, as considered in this paper, it is fully described by a vector $p \in [0,1]^{\mathbb{N}}$. One natural estimate for $p$ is the empirical mean. In previous work, the performance of this estimator (in terms of the expected $\ell_\infty$ error) was tightly characterized. In particular, the class LGC (local Glivenko-Cantelli) of those $p$ for which this error vanishes as the sample size $n$ grows is tightly characterized. This work asks how this landscape changes if one allows alternative estimators (the answer: not too much).

In particular, if a class $\mathcal{P}$ satisfying some rather mild regularity conditions is learnable by any estimator, they prove that $\mathcal{P}$ is essentially a subset of LGC (up to a minor, natural tweak to this class). They also provide counter examples showing that these regularity conditions cannot be fully removed (although there are some interesting questions about relaxing them a bit). Moreover, they prove a minimax lower bound showing that the risk bound achieved by the empirical mean estimator cannot be improved too much (for any estimator).

## Update after Rebuttal
I maintain my positive score.

**Claims And Evidence:**

The proof of Thm 2.1 uses some careful applications of Neyman-Pearson and a lemma from a previous paper in this area. I did not dig into the previous proof of that lemma, but the argument otherwise looks sound.

The proof of Thm 2.2 begins with a natural construction and application of Fano's inequality. However, the exact tuning of the parameters requires a lot of care, and I think this could be better explained. I followed the individual steps of the proof, and, assuming that cited results are true, it appears sound. However, I would feel more comfortable vouching for correctness if there was more high-level discussion about parameter tuning.

I think there are some easily fixed issues with the proof of Thm 2.3, which I mention below. But I am sure the result is true.

Proposition 2.4 has a direct and complete proof.

**Essential References Not Discussed:**

n/a

**Experimental Designs Or Analyses:**

There are a few supporting plots in the Appendix. At a skim, they look good (but even if there were issues I do not think they play a critical role).

**Methods And Evaluation Criteria:**

n/a

**Other Comments Or Suggestions:**

On page 4, bottom right, there are some missing indices from the $p^{(Y)}$ vector on the second and third lines.

On page 7, in the first equation block of Step 1, $B_j$ should be $E_j$.

**Other Strengths And Weaknesses:**

The paper is generally well written and nice to read.

As someone familiar with the methods in this work and some of the earlier results on LGC, I have a bit of a hard time assessing significance of this paper's results. Are there any interesting applications of LGC to other statistical / learning theoretic problems? Also, I am not viewing any of the proof techniques as primary contributions in their own right. If there is any major technical difficulty to one of the proofs that would rule out simpler approaches, this should be highlighted.

The statement of Thm 2.2 is a bit funny. It seems that first string of inequalities on $n,s,t$ can be removed, since they appear again right afterwards. I think some further discussion is warranted on the gap between this LB and the UB. I also think the proof could use some more high-level discussion, as mentioned above.

**Questions For Authors:**

Suppose I am interested in mean estimation, rather than distribution estimation. What does known estimation landscape look like there beyond product distributions, and would any of your results translate?

**Relation To Broader Scientific Literature:**

The study of the local Glivenko Cantelli class was initiated pretty recently and this paper cites the relevant work that I know if, though I am not an expert on this specific problem. The existing work focused on the empirical mean estimator, whereas this work considers the limits of learnability using any estimator.

**Theoretical Claims:**

Yes, as discussed above.

In the proof of Thm 2.3, the estimator defined at Step 3 should involve a sum over $\hat{p}_n(i)$ rather than $\hat{p}_n(j)$, I believe. Moreover the discussion at Step 4 is of convergence for fixed $j$, when the question is really about uniform convergence over $j$. Without the need for a uniform guarantee there is no need to use the alternate estimator. Of course, it is clear that the authors understand this, and it is easy to fix.

---

> ### Author Rebuttal · Authors · 2025-03-31
>
> Thank you for your constructive review. We appreciate your insights and address your key concerns below:
>
> 1. Parameter Tuning and Proof Clarity (Theorems 2.2 and 2.3):
>    We recognize that parameter tuning in Theorem 2.2 is indeed the main challenge. We intend to offer clearer intuition on how these parameters are selected and why they yield the stated minimax bound. In Theorem 2.3, we will correct the minor issues concerning the estimator’s definition at Step 3 (e.g., using $\hat{p}^n(i)$ instead of $\hat{p}^n(j)$) and refine the discussion on uniform convergence to avoid confusion regarding a fixed $j$.
>
> 2. Significance and Applications:
>    We understand your request for a broader context of potential applications of LGC. While our primary focus is the Local Glivenko-Cantelli framework, we will highlight possible connections to mean estimation in correlated or more complex settings. Where relevant, we will also discuss why simpler approaches do not suffice, underscoring the nontriviality of our techniques.
>
> 3. Minor Corrections and Notational Consistency:
>    We will fix the missing indices in the expression for $p^{(Y)}$ (page 4, bottom right) and replace $B_j$ with $E_j$ on page 7 (Step 1), ensuring accuracy and consistency throughout.
>
> 4. Beyond Product Distributions:
>    As for extending these findings to mean estimation in non‐product settings, note that $\ell_\infty$ mean estimation is not widely studied beyond the references cited in our introduction, and we are not aware of additional relevant work.
>
> We appreciate your thoughtful review and hope these clarifications address your concerns.

---

### Official Review · Reviewer_TTqR · 2025-03-19

**Overall Recommendation:** 3

**Summary:**

The paper discusses the "Local Glivenko Cantelli" problem focusing on families of product distributions. There are three main results:

The first main theorem (Theorem 2.1) argues that LGC (the family of product measures that is learnable by the empirical mean estimator EME) is the largest family learnable by any fixed estimator, by showing any family of product measures that decays, is strongly symmetric about 1/2, and is learnable belongs to LGC.

The second main theorem (Theorem 2.2) argues that EME is nearly minimax-optimal by establishing a minimax lower bound for any estimator based on n i.i.d. samples from the latent distribution and comparing it with established bounds from results in the literature.

The third main theorem (Theorem 2.3) shows that LGC can be expanded if the learner can exploit structural knowledge of the problem. In these scenarios, some assumptions about decay and symmetry may be relaxed.

**Claims And Evidence:**

The paper is mostly theoretical, and the claims are clear. There are some questions about the proofs that are stated in a later section of the review. The paper explains the main results clearly and provides proof for each of the main theorems.

**Essential References Not Discussed:**

N/A.

**Ethical Review Concerns:**

N/A.

**Experimental Designs Or Analyses:**

There are not many experiments provided in the paper. It would be helpful to give a description of the simple average estimator, which was not included in the paper although it was used extensively to compare against the EME.

**Methods And Evaluation Criteria:**

Since the paper is mainly theoretical, there are limited discussions about empirical methods or evaluation criteria. Some simulations are provided for the EME and the "simple average estimator", showing that the latter performs better, while both illustrate tightness of minimax bounds presented in Theorem 2.2.

**Other Comments Or Suggestions:**

One minor typo:

1. Line 145 on the right side: "Theorems 2.1".

**Other Strengths And Weaknesses:**

Overall, I found the paper to be a nice read. The authors did a great job describing the problem setting and presenting the main results.

One small concern is that the main results are quite short -- descriptions, main results, and discussions were mostly within the first three pages, while the remaining main text were dedicated to proofs of main theorems. Considering there have been multiple papers in the literature on LGC, I think it would be helpful if the paper contains a section highlighting novelties in proof techniques that this paper introduces. It would also be great if the paper could contain some discussions of applications or implications to real-world scenarios.

**Questions For Authors:**

I have some questions related to proofs:

**Proof of Theorem 2.1**

1. I did not fully understand why the assumption $\dot{p}_j^*\in[0,1/4]^\mathbb{N}$ incurs no loss of generality, especially when it is used to lower bound the minimax risk later.
2. It might be worthwhile to elaborate what optimality criteria the Neyman-Pearson lemma gives to an estimator. For instance, how do we guarantee the optimal estimator $\hat{y}$ that minimizes the posterior probability of error also minimizes the expected $L_\infty$ norm? It would be helpful to clarify these since the argument of the contradiction builds heavily on analysis of the posterior probability.

**Proof of Theorem 2.2**

3. In line 260 on the left side, why is $\|\|p^{(k)}-p^{(\ell)}\|\|_\infty = |q'-q|$? Are we missing an assumption that $q'\geq 2q$? A similar statement is also used later around line 283-284 on the left side.

**Proof of Theorem 2.3**

4. The definition of the test function is a bit ambiguous. Is it safe to presume $B_j$ in line 355-356 on the right side should be $E_j$?

5. It would be helpful to provide statements referenced from literature. For instance, in Step 2., Lemma 3 of Cohen & Kontorovich 2023 are repeatedly referenced. It would be helpful to state the lemma.

**Relation To Broader Scientific Literature:**

Based on the descriptions in the paper, there were multiple related problem settings in the literature that have been explored with fruitful results. This paper identifies a new direction but borrows many tools and techniques from the literature.

**Theoretical Claims:**

I read the proofs for all three theorems but did not check all proofs rigorously. Some questions are listed below in the "questions for authors" section.

---

> ### Author Rebuttal · Authors · 2025-03-31
>
> We thank you for your detailed and insightful evaluation and for your positive comments regarding the clarity of our main results and the rigor of our proofs. Below, we address your specific concerns:
>
> 1. Regarding the assumption $\dot{p}_j^*\in[0,1/4]^\mathbb{N}$, note that we can disregard any values where $\dot{p}_j > 1/4$ without loss of generality. In fact, excluding these values only serves to decrease the supremum in our minimax analysis, which we have shown is lower bounded by 1/4. We will clarify this point further.
> 2. We acknowledge the importance of the optimality criteria provided by the Neyman-Pearson lemma. In our current draft, we refer to the lemma without additional discussion; we will expand on its role to clarify how minimizing the posterior error probability aligns with minimizing the expected $L_{\infty}$ norm.
> 3. Concerning the inequality in Theorem 2.2, there is no need to assume $q'\geq 2q$. The equality $\|p^{(k)}-p^{(\ell)}\|_\infty = |q'-q|$ follows directly from our construction, as it represents the maximum coordinate-wise difference between $p^{(k)}$ and $p^{(\ell)}$.
> 4. We agree that the notation is inconsistent; $B_j$ should indeed be $E_j$, and this will be corrected.
> 5. We will also include an explicit statement of the referenced lemma from Cohen & Kontorovich (2023) to ensure clarity.
>
> Additionally, we appreciate your suggestion to provide a description of the simple average estimator used in our simulations. Finally, we note your inquiry regarding the broader significance of our work; our study builds upon a well-established framework with direct implications in learning theory, and the technical challenges we address are of independent interest.
>
> We thank you again for your constructive feedback.

---

### Official Review · Reviewer_Frhk · 2025-03-21

**Overall Recommendation:** 4

**Summary:**

This paper investigates the mean estimation problem in the binomial empirical process. First, under mild technical conditions, it establishes that the LGC class, as defined in Cohen & Kontorovich (2023), is the largest class that is learnable by any estimator. Furthermore, it demonstrates that the empirical mean estimator (EME) achieves the minimax optimal rate. Finally, the paper provides examples of learnable classes that do not require a decaying assumption, broadening the scope of learnability beyond previously studied settings. Overall, this work offers a more comprehensive understanding of mean estimation in the binomial empirical process.

**Claims And Evidence:**

Yes. The claim is clear and well-supported by theorems and proofs.

**Essential References Not Discussed:**

No

**Experimental Designs Or Analyses:**

Does not apply.

**Methods And Evaluation Criteria:**

Does not apply.

**Other Comments Or Suggestions:**

No

**Other Strengths And Weaknesses:**

The paper is well-written, and the contribution is solid.

The main limitation is that the problem is somewhat standalone and lacks strong connections to broader machine-learning topics. The paper could be strengthened by exploring its relationship with general empirical process theory and highlighting its relevance.

**Questions For Authors:**

No

**Relation To Broader Scientific Literature:**

The problem is a bit standalone and a bit disconnected from broader machine learning literature.

**Theoretical Claims:**

This is a theory paper. While I did not verify every detail of the proofs, they appear sound at a glance. The contribution is solid.

---

> ### Author Rebuttal · Authors · 2025-03-31
>
> Thank you for your thorough and encouraging evaluation. We appreciate your recognition of the soundness and clarity of our theoretical contributions and proofs. Regarding the observation that the problem might appear somewhat standalone, we would like to emphasize that the original LGC paper by Cohen & Kontorovich (2023) provides a detailed discussion on its broader connections to machine learning, notably illustrating how the problem emerged from an analysis of the multiplicative weights algorithm. Uniform convergence in the Glivenko-Cantelli sense remains a cornerstone for deriving generalization bounds, which underscores the relevance of our investigation to the wider ML community.
>
> > The main limitation is that the problem is somewhat standalone and lacks strong connections to broader machine-learning topics. The paper could be strengthened by exploring its relationship with general empirical process theory and highlighting its relevance.
>
> In the revision, we will elaborate upon the motivation and better situate the paper in the context of recent developments.

---

### Official Review · Reviewer_cB2y · 2025-03-22

**Overall Recommendation:** 3

**Summary:**

This paper focus on the Local Glivenko Cantelli setting, which studies the uniform convergence rates of Empirical Mean Estimator (EME).

**Claims And Evidence:**

Yes.

**Essential References Not Discussed:**

No.

**Experimental Designs Or Analyses:**

No.

**Methods And Evaluation Criteria:**

Yes.

**Other Comments Or Suggestions:**

No.

**Other Strengths And Weaknesses:**

While I appreaciate the technical rigor, I believe the main paper should focus on the intuition and logical development instead of technical proofs.

**Questions For Authors:**

No.

**Relation To Broader Scientific Literature:**

It should be worthwhile to focus on the key questions without introducing much technicality in the introduction section to make the topic more appealing to the broader scientific literature.

**Theoretical Claims:**

No.

---

> ### Author Rebuttal · Authors · 2025-03-31
>
> Thank you for your constructive feedback. We fully agree that presenting the key questions in an intuitive way is valuable. In our work, we carefully balanced the need to convey the underlying ideas with the necessity of rigorous technical proofs, given the inherent complexity of the Local Glivenko-Cantelli setting. We did strive to emphasize the logical development and core insights, but simplifying the exposition further proved challenging without compromising the correctness of our results.
>
>
> > While I appreaciate the technical rigor, I believe the main paper should focus on the intuition and logical development instead of technical proofs.
>
> In the revision, we will be happy to expand upon the intuition and logical development.

---

### Decision · Program_Chairs · 2025-05-01

**Decision:**

Accept (poster)

**Comment:**

As described by the reviewers, the paper presents new important results for the estimation of the mean in binomial empirical processes. The paper provides a significant theoretical contribution for the study of uniform convergence of mean estimation.

Some reviewers have pointed out the somehow unclear connection with a broader machine learning community. I would like to encourage the authors to further describe the relevance of the results presented for general machine learning methods, in particular for more general empirical processes. In that sense, the authors can update the paper based on the discussions with the reviewers during the rebuttal.